# Factors Associated with COVID-19 Vaccine Acceptance among Healthcare Professionals and Community Stakeholders in Hong Kong: A Cross-Sectional Study

**DOI:** 10.3390/ijerph192114499

**Published:** 2022-11-04

**Authors:** Regina Lai Tong Lee, Wai Tong Chien, Michelle Stubbs, Winnie Lai Sheung Cheng, Daniel Cheung Shing Chiu, Keith Hin Kee Fung, Ho Yu Cheng, Yuen Yu Chong, Anson Chui Yan Tang

**Affiliations:** 1The Nethersole School of Nursing, Faculty of Medicine, The Chinese University of Hong Kong, Hong Kong; 2School of Nursing and Midwifery, College of Health, Medicine and Wellbeing, The University of Newcastle, Callaghan 2308, Australia; 3School of Nursing, Tung Wah College, Hong Kong; 4Department of Paediatrics, Faculty of Medicine, The Chinese University of Hong Kong, Hong Kong; 5School of Nursing, The Hong Kong Polytechnic University, Hong Kong

**Keywords:** factors, vaccine acceptance, healthcare professionals, community stakeholders

## Abstract

Background: Acceptance of vaccination in both healthcare professionals and the general public in the community is vital for efficacious control of the virus. Vaccine acceptance associates with many factors. Little research has been dedicated to examining attitudes and behaviors of healthcare professionals and community stakeholders regarding COVID-19 vaccine acceptance in Hong Kong. Methods: An online cross-sectional survey was sent between February and April 2021 (N = 512). Multivariable regression modeling was used to identify associated variables with outcomes using adjusted odds ratios (AOR) and 95% of confidence intervals (CI). Results: Two demographic variables—age group of over 40 years old (40–59: ORm = 3.157, 95% CI = 2.090–4.467; 60 or over: ORm = 6.606, 95% CI = 2.513–17.360) and those who had previously received a flu vaccination (ORm = 1.537, 95% CI = 1.047–2.258)—were found to be associated with high vaccine intent. Adjusting for these two variables, the results showed that five factors on knowledge variables as perceived benefits for vaccine intent were statistically significant: “Closed area and social gathering are the major ways of SAR-CoV-2 transmission” (AOR = 4.688, 95% CI = 1.802–12.199), “The vaccine can strengthen my immunity against COVID-19, so as to reduce the chance of being infected with it” (AOR = 2.983, 95% CI = 1.904–4.674), “The vaccine can lower the risk of transmitting the viruses to my family and friends” (AOR = 2.276, 95% CI = 1.508–3.436), “The benefits of COVID-19 vaccination outweigh its harm” (AOR = 3.913, 95% CI = 2.618–5.847) and “Vaccination is an effective way to prevent COVID-19” (AOR = 3.810, 95% CI = 2.535–5.728). Conclusions: High vaccine intent was associated with age and having previously received a flu vaccination. Knowledge and attitudes of healthcare professionals and community stakeholders were associated with high vaccine intent. Training and continuing education programs for healthcare providers and community stakeholders focusing on the delivery of evidence-based data on the benefits of vaccination campaigns for populations to increase the vaccination rates is recommended.

## 1. Introduction

Protection against COVID-19 through vaccination is not merely dependent on vaccine efficacy and safety. The acceptance of vaccination among both healthcare professionals and the public in the community is vital for the efficacious control of the virus [1,2]. It is also commonly known that healthcare providers are at a high risk of exposure to COVID-19 compared with the general population. This increased risk also acts as a potential threat to their own health and the health of patients [3]. As the COVID-19 pandemic has been continuing to unfold globally since 2019, the number of cases of COVID-19 is still rapidly increasing in many countries. As of 9 June 2022, there have been 530,896,347 cumulative cases of COVID-19, including 6,301,020 deaths, reported to the World Health Organization (WHO) [4]. In Hong Kong, 1,216,842 cumulative cases of COVID-19 have been reported, including 9390 deaths [5]. Vaccination remains an important component of preventive care for the outbreak of the COVID-19 disease globally. Vaccine acceptance and hesitancy have been influenced by social, communication, practical and cultural factors such as trust, fear and anxiety, safety, information sufficiency, conspiracy beliefs and social influence [6]. These prevailing factors were commonly known as barriers to achieving herd immunity [7]. A lack of trust within certain community population groups may arise from previous devastations in the healthcare systems and public institutions leading to a lack of vaccine acceptance [2]. Thus, it is important to investigate the perceptions and those associated factors influencing the healthcare providers’ and community stakeholders’ acceptance of the COVID-19 vaccine [1]. Vaccine hesitancy is considered by the WHO as one of the top ten threats to global health [8]. Although community vaccination campaigns and programs are essential to evaluate the effectiveness of the injection of vaccines, mass vaccination against COVID-19 of the populations around the world creates an enormous challenge for the governments to build public trust in accepting the COVID-19 vaccines [9].

Public willingness to accept newly promoted vaccines vary with social class, time, ethnicity and contextual human behavioral factors [7]. In this regard, frequent communications between healthcare professionals and different population groups is critical to address the hesitancy-associated predictors and to motivate vaccine-hesitant individuals towards vaccine acceptance. Thus, healthcare professionals and community stakeholders play an important role in delivering up-to-date information and data to the people at large to foster vaccine confidence and to encourage people to be vaccinated willingly. Many countries are still in the planning phase, ironing out health education activities and health promotion campaigns surrounding the fourth dose of the COVID-19 vaccination. Despite the proven positive impacts of vaccination programs in prevention, the elderly aged over 60 years old had the lowest vaccination rate and parents were unwilling to have their children vaccinated in Hong Kong [10,11]. There is a need to obtain additional data on barriers and facilitators of the uptake of COVID-19 vaccination. This includes the views and opinions on the uptake or delay of vaccinations from primary professional healthcare providers (mainly nurses, doctors, healthcare assistants and social workers) and community stakeholders (including policy makers, community leaders, school principals, schoolteachers and parents) [12].

Therefore, this study aimed to: (1) investigate the relationships between knowledge, attitudes and behaviors among healthcare professionals and community stakeholders towards COVID-19 vaccine acceptance; (2) identify the associated factors with healthcare professionals’ and community stakeholders’ knowledge, attitudes and behaviors towards the COVID-19 vaccine uptake.

## 2. Methods

### 2.1. Study Design and Participants

This was a cross-sectional online survey study from February to April 2021. Convenience snowball sampling was used to recruit study participants via various online platforms, including online seminars, professional associations’ websites, Facebook, Twitter and community organization websites in Hong Kong. Inclusion criteria included: (1) aged 18 years or above; (2) able to read and write the Chinese language; (3) able to give consent; (4) currently working as either a full-time or part-time healthcare professional or being identified by the researcher as a community stakeholder. A total of 520 healthcare professionals and community stakeholders returned the online survey with implied consent forms via the accessed link sent to them; however, data from 512 study participants were analyzed. Eight study participants were excluded due to not completing all components of the online survey. The study was approved by the Institutional Review Committee of the Hong Kong Nurses’ Association in Hong Kong.

### 2.2. Measure

The study instrument consisted of two components: (1) a demographic sheet (46 items, of which 8 items detailed personal characteristics (items 37–44)); (2) modified from a Chinese version of the community stakeholder vaccination attitude scale (C-CSVAS) (38 items) [13].

Socio-demographic factors included gender (male and female), age groups (<40 years old, 41–59 years old, >60 years old), ethnicity (Chinese vs. non-Chinese), education, employment, job experiences, types of job and ranking, included as personal characteristics.

The C-CSVAS comprises four subscales: 22 items (No. 1–22) on up-to-date knowledge of side effects of each vaccine and own health conditions fit for vaccination, 6 items (No. 23–28) on attitude, 4 items (No. 29–32) on intention, 2 items (No. 33–34) on action and 2 items (No. 45–46) on the uptake of the COVID-19 vaccination [13]. The original C-CSVAS was utilized in a previous study that aimed to explore the knowledge, attitudes, intentions and actions of the uptake of the influenza vaccination among hospital and community healthcare providers and community stakeholders in Hong Kong [13]. Two items were added to the instrument, aiming to investigate the confidence and support provided by government and health services in managing the pandemic (No. 35–36).

The four subscales of the C-CSVAS, which had been modified from a previous study with the content validity index of 0.79, was obtained from a group of five panel members, including an infectious control clinician, a medical doctor and three academic staff. Cronbach’s alpha ranged from 0.78 to 0.83 for the C-CSVAS [13]. The four refined subscales included: (1) up-to-date knowledge of the vaccine (side effects and own health conditions); (2) attitudes (perceived barriers of vaccines and worries about unforeseen future effects); (3) acceptance or intention (perceived benefits of vaccines); (4) action (vaccine intent, for example, to achieve herd immunity) for the uptake of the COVID-19 vaccination.

The scoring or rating of subscales are as follows: (1) first and second subscales (knowledge and attitude): yes, no and I do not know; (2) third subscale (acceptance): strongly disagree, disagree, agree, strongly agree; (3) fourth subscale (action and intention): two choices for the uptake of the influenza and the COVID-19 vaccination with the “yes or no” rating.

Uncertainty and unwillingness to vaccinate against COVID-19 when available were based on one item: “How likely to do you think you are to get a COVID-19 vaccine when one has been approved?” Response options ranged from “1 strongly disagree” to “6 strongly agree”.

Responses to online survey questions with an ordinal variable for the positive and negative factors against COVID-19 vaccination were coded: (0) intend to vaccinate (responses of 4–5) and (1) unsure about whether to vaccinate (responses of 1–3). Two binary variables were created to compare individuals who have a positive attitude versus a negative attitude between the healthcare providers either working in the hospital or community settings and between the healthcare providers versus the community stakeholders.

Responses to the questions on compliance with government COVID-19 guidelines are on a scale from 1 (none at all) to 7 (very much so). We analyzed this as a binary variable reflecting higher (6–7) vs. lower (1–5) compliance. Knowledge of COVID-19 was measured with the questions and rated on a 3-point scale from 1 (very poor knowledge) to 3 (very good knowledge). Responses of 1–4 were categorized as very poor/poor compared with very good/good (5–7) COVID-19 knowledge. 

The presence or absence of having been infected with COVID-19 was categorized as a binary variable (“yes, diagnosed and recovered”; “yes, diagnosed and still ill”; or “not formally diagnosed but suspected” versus “no, not that I know of” or “I don’t know”). Prior vaccine behaviors were based on two yes/no questions. 

Responses to the two last questions of the C-CSVAS on the confidence and support provided by government and health services to handle the pandemic were assessed with one question each.

Between February and April 2021, the online survey was posted (with informed consent) via Google Forms. Participants were asked to complete an online survey developed in Google Forms. The online survey commenced and consent to participate in the study was considered valid once participants had checked the box of the statement: “I read through the information above and agree to participate in the study. I can stop anytime if I do not want to continue.” This statement was located in the middle of page one of this anonymous online survey. The participants took approximately 15–20 min to complete this online survey.

### 2.3. Statistical Analysis

Regarding the reliability of the online survey, the value of Cronbach’s α was performed to check its internal consistency. Descriptive statistics were used to analyze data for a single categorical variable including means, frequencies and percentages. A multiple stepwise regression model was computed to explore the linear relationship between the explanatory independent variables (demographics, knowledge, positive and negative factors) and dependent variable (intention to get vaccinated). The independent variables include demographics such as age and previously received flu vaccines, knowledge and positive and negative factors; multivariate odds ratios (ORm) were therefore derived. Associations between the independent variables (demographics, knowledge, attitudes) one by one and the dependent variable (intention to get vaccination) were assessed, adjusting for those background variables that were found to be significant in the multivariate analysis; adjusted odds ratios (AOR) were then derived. Corresponding 95% confidence intervals (CI) of odds ratios were presented. Statistical analyses were performed using the statistical software SPSS 24 with *p* values < 0.05 taken as statistically significant.

## 3. Results

### 3.1. Characteristics of Participants

In Table 1, a total of 512 participants were included in the analysis, of whom 83.8% were female. A total of 58.0% were engaged in health-related jobs and half of them (53.1%) were aged below 40 years. Nearly half of the participants (52.1%) had a higher risk of being infected at work or in daily life and most participants (88.7%) did not have chronic illness. Most family or friends of the participants were not admitted to hospital due to a COVID-19 infection. Nearly half of the participants (47.5%) had received an annual flu vaccination in the past 3 years (Table 1). Six surveys with missing data were excluded.

Regarding reliability of the online survey, the values of Cronbach’s α for knowledge of COVID-19 (Q1–Q22), factor against COVID-19 vaccination (Q23–Q28) and positive factors for COVID-19 vaccination (Q29–Q36) were 0.696, 0.692 and 0.843, accordingly [14]. These are acceptable values for the reliability of the study questionnaire. The prevalence of intention to take (acceptability) COVID-19 vaccines was 65.0%. Regarding the knowledge about COVID-19 transmission, the prevalence of appropriate responses for individual knowledge items was high, ranging from 72.5% to 97.9% (Table 2). Most (97.9%) of the participants perceived that COVID-19 was highly contagious. Most participants reported insight on effective ways to mitigate COVID-19, including being aware of hand hygiene, wearing facial masks and practicing social distancing (99.0%), and that the virus could be transmitted to family members or friends if one was diagnosed with COVID-19 (98.2%). In contrast, there was a misconception that youths have a higher risk of getting COVID-19 (11.9%).

### 3.2. Healthcare Professionals’ and Community Stakeholders’ Knowledge and Attitudes towards COVID-19 Vaccine Acceptance 

Regarding up-to-date knowledge about the COVID-19 virus itself, the prevalence of appropriate responses for individual knowledge items ranged from 46.1% to 97.9% (Table 2). Most of the participants perceived that COVID-19 could cause serious and life-threatening complications in people with chronic illnesses (97.9%) and in children (85.7%). Many also perceived that those who recovered from COVID-19 may get infected with it again (94.5%) and that there was no way to effectively cure COVID_19 currently (79.5%). In contrast, 39.8% had a misconception that a majority of young and healthy people would not have complications even if they suffer from COVID-19 (Table 2).

Regarding knowledge about the effect of the COVID-19 vaccine, most participants perceived that vaccination could strengthen immunity against COVID-19, thus preventing people from being infected (77.5%). Most participants perceived that the vaccine could lower the risk of virus transmission to family members and friends (70.3%) and that it was an effective way to prevent COVID-19 (66.6%). Many also believed that it did not have 100% protection (79.9%), did not have a lifelong protection after completing two doses of vaccine (69.1%) and that the benefits of the vaccination outweighed its harm (58.6%). The prevalence of an appropriate response for individual knowledge items ranged from 58.6% to 79.9% (Table 2).

Concerning attitudes towards the acceptance of the COVID-19 vaccine, the prevalence of responses reflecting perceived barriers of COVID-19 vaccines were: (1) unknown side effects (89.1%); (2) insufficient scientific evidence to prove its effectiveness (73.2%); (3) confusing information about the vaccine and difficulties in selecting which vaccine is suitable (69.3%); (4) minimal chance of getting COVID-19 because all the preventive measures had been done (50.4%). Concerning perceived benefits, the prevalence of agreement or strong agreement with the statements was: “It will encourage me to get the vaccination if there is sufficient scientific evidence to support the effectiveness of the vaccines and show no severe side effects” (90.0%); “The government and social media should provide accurate and clear information and instruction for the vaccine to the general public” (90%); “The government should set up a trust fund to provide assistance to those experiencing severe side effects after vaccination” (90%); “If more family members or friends get the vaccination without undesirable effects, I will consider the vaccination” (69.9%); “The general public has the responsibility to get the vaccination to reduce COVID-19 transmission” (68.4%); “If the vaccination venue and time fit me better, it will increase my intention of vaccination” (57.8%); “It will encourage me to get the vaccination if the vaccination is recommended by prestigious healthcare professionals” (58.4%) (Table 2).

### 3.3. Factors Associated with Healthcare Professionals’ and Community Stakeholders’ Knowledge and Attitudes towards COVID-19 Vaccine Intent Behaviors

A regression model analysis was conducted. Regarding factors associated with healthcare professionals’ and community stakeholders’ knowledge and attitudes towards vaccine intent behaviors, two of the background variables that are listed in Table 3 were statistically significant: aged 40 and above (40–59: ORm = 3.157, 95% CI = 2.090–4.467; 60 or over: ORm = 6.606, 95% CI = 2.513–17.360) and those who had received the flu vaccination in the past 3 years (ORm = 1.537, 95% CI = 1.047–2.258). Adjusting for these two variables, the results showed that five factors on knowledge variables as perceived benefits for vaccine intent were statistically significant: “Closed area and social gathering are the major ways of SAR-CoV-2 transmission” (AOR = 4.688, 95% CI = 1.802–12.199), “The vaccine can strengthen my immunity against COVID-19, so as to reduce the chance of being infected with it” (AOR = 2.983, 95% CI = 1.904–4.674), “The vaccine can lower the risk of transmitting the viruses to my family and friends” (AOR = 2.276, 95% CI = 1.508–3.436), “The benefits of COVID-19 vaccination outweigh its harm” (AOR = 3.913, 95% CI = 2.618–5.847) and “Vaccination is an effective way to prevent COVID-19” (AOR = 3.810, 95% CI = 2.535–5.728). Those who perceived that the vaccine could strengthen immunity to reduce the chance of being infected were about thrice more likely to get vaccinated than those who did not. Those who perceived the vaccine could lower the risk of transmitting the virus to family members or friends, those who perceived that the benefits of COVID-19 vaccination outweighed its harms and those who perceived that vaccination was an effective way to prevent COVID-19 tended to be more likely to get vaccinated.

Three positive and negative factors relating to knowledge variables as perceived barriers for vaccine acceptance were significant: In Table 4, “The vaccine is still under development. There is insufficient evidence to prove its effectiveness against COVID-19” (AOR = 0.481, 95% CI = 0.298–0.775), “You are not within high-risk groups. You can fully recover from the illness with no complications” (AOR = 0.594, 95% CI = 0.385–0.917) and “You have minimal chance of getting COVID-19 because you have done all the preventive measures” (AOR = 0.567, 95% CI = 0.386–0.834). This implies that those who had less confidence in the effectiveness of the vaccine, those who perceived that they were not within high-risk groups and those who believed that they had less chance of being infected were 51.9% ((1 − 0.481) × 100%), 40.6% ((1 − 0.594) × 100%) and 43.3% ((1 − 0.567) × 100%), accordingly, less likely to get vaccinated than those who believed in the opposite.

In Table 4, four positive factors regarding behavioral variables as perceived benefits for vaccine acceptance were significant: “If more family members or friends get the vaccination without undesirable effects, I will consider the vaccination” (AOR = 3.714, 95% CI = 2.447–5.636), “If the vaccination venue and time fit me better, it will increase my intention of vaccination” (AOR = 4.348, 95% CI = 2.895–6.529), “It will encourage me to get the vaccination if the vaccination is recommended by prestigious healthcare professionals” (AOR = 2.377, 95% CI = 1.602–3.528) and “It will encourage me to get the vaccination if there is sufficient scientific evidence to support the effectiveness of the vaccines and show no severe side effects” (AOR = 4.653, 95% CI = 2.389–9.063).

Other factors leading to a higher vaccination rate would be issuing a vaccination mandate for all high-risk groups, having a trust fund to help with people experiencing severe side effects after vaccination and promoting the notion that taking the vaccine is a public responsibility. In Table 4, “The general public has the responsibility to get the vaccination to reduce COVID-19 transmission” (AOR = 5.059, 95% CI = 3.309–7.735), “The government should mandate all high-risk groups to get the vaccination” (AOR = 2.235, 95% CI = 1.467–3.404) and “The government should set up a trust fund to aid those experiencing severe side effects after vaccination” (AOR = 1.858, 95% CI = 1.003–3.440).

## 4. Discussion

The study findings reported that most participants perceived that knowledge of the vaccine, and its perceived benefits could strengthen the immunity of participants via the uptake of vaccination against the COVID-19 disease, preventing people from being infected in the communities. A majority of participants perceived that the vaccine could lower the risk of transmitting the virus to family members and friends and that it is an effective way to prevent COVID-19 outbreaks. For those who perceived that close areas and social gatherings were the major ways of SAR-CoV-2 transmission were 4.688 times more likely to get vaccinated than those who did not. This finding is different from a study conducted in Australia, Singapore and Hong Kong that reported that increased knowledge was negatively associated with high vaccine intent [15]. The study findings also identified other prevailing factors such as perceived benefits and barriers towards the COVID-19 vaccine, along with acceptance and intent associated with study participants’ knowledge and attitudes.

Concerning the variable attitude, study participants believed that “harm” outweighed the “benefits” of vaccination if: (i) you did not have full protection and (ii) you did not have a lifelong protection after completing two or three doses of the vaccine. This finding is similar to a study conducted in Hong Kong that investigated the perceived severity of the pandemic, the perceived benefits of the vaccine, the perceived access and barriers and harm and trust in the healthcare system and vaccine manufacturers that were all associated with COVID-19 vaccine acceptance among the working population, including clerical/services/sales workers [6].

In this study, two demographic variables, including age group over 40 years old and those who had previously received a flu vaccination, were found to be associated with high vaccine intent. Our study findings are similar to an Indian study that reported people aged 45 years or older were more agreeable to the uptake of COVID-19 vaccines [16]. Furthermore, a study conducted in Ethiopia, identified that age and profession were significantly associated with health professionals’ attitudes towards the COVID-19 vaccine intent [1].

Regarding vaccine-related information for the public such as trust in the vaccine manufacturers and the government’s communications/reports about the impact of the vaccine on people’s health status has been identified as one of the key factors influencing participants’ opinions towards COVID-19 vaccine acceptance. Our study findings report that participants are more likely to get vaccinated if there is evidence that the vaccine is effective in preventing COVID-19 and would not cause any severe side effects. This finding is similar to the other literature that report “transparency in reporting the number of newly diagnosed COVID-19 cases and the deaths is mandatory as these factors are the main determinants of COVID-19 vaccine acceptance”. A systematic review identified 11 potential factors influencing COVID-19 vaccine acceptance and hesitancy and it revealed that people would connect with informative and effective messaging that clarified COVID-19 vaccine safety, side effects and effectiveness [17].

Regarding the uptake of the vaccine injection, we report that the study participants agreed that it is a public responsibility as a citizen of Hong Kong. We found that most study participants agreed that to lead to a higher vaccination rate it would be necessary to issue a vaccination mandate for all high-risk groups to fulfil the role of a citizen in Hong Kong. This finding is supported by a published article that suggests that educational interventions: (1) highlight the benefits of vaccination; (2) address public safety issues; (3) agree that it is the individual’s social responsibility to reach herd immunity and to control the pandemic [18]. Another significant finding on the attitudes of vaccinated healthcare professionals and community stakeholders is supported by a previous study that examined medical students’ attitudes towards vaccination following curricular intervention [19]. The medical students’ personal experiences, knowledge and skills associated with counseling and administration of the vaccine were associated with their acceptance of the vaccination injection. The findings of this study also reinforced that healthcare professionals with in-depth knowledge about the vaccine and those healthcare professionals with positive attitudes about the vaccine were more likely to recommend vaccination to family and friends and felt more comfortable counseling about the vaccine [7].

In addressing disparities in access to obtaining a free COVID-19 vaccine, this study found that participants expressed their willingness to undergo vaccination if the venue (place of vaccination) is convenient to them and if their scheduled time for vaccination is flexible. Having a selection of accessible venues and flexible timeslots, in addition to being free of charge, could increase rates of vaccination uptake, especially for those low-socioeconomical high-risk groups. This finding is further supported by a study conducted in Hong Kong about foreign domestic workers with low literacy failing at COVID-19 safeguards and unintentionally sharing this illness largely within the communities [20].Another study, conducted in Ghana and Bangladesh, reported on barriers to vaccination uptake (including vaccine cost and accessibility) that would arise when the rollout of the COVID-19 vaccination occurred in lower-middle income countries [21]. Public education campaigns and recommendations of healthcare professionals have adequately addressed vaccine hesitancy previously. Unique partnerships between governments and major players, including healthcare professionals and stakeholders, will guarantee the effective promotion of vaccination programs and campaigns within communities as recommended by the World Health Organization (WHO) and the Organisation for Economic Co-operation and Development (OECD) [9]. This unique partnership may increase vaccination uptake, as vaccine acceptance and hesitancy had been flagged by the WHO as one of the top 10 threats to global health [22].

This study aimed to investigate and understand healthcare professionals’ and community stakeholders’ knowledge, attitudes, behaviors and the associated factors towards COVID-19 vaccine acceptance. As localized vaccine programs and campaigns are beneficial to counter misinformation distributed throughout community populations in United Kingdom [23], it is important to understand the healthcare professionals’ and community stakeholders’ attitudes and their intention for the uptake of the COVID-19 vaccination program launched by the health authority. Future research should include a variety of roles and responsibilities of healthcare professionals and should understand their views and perceptions, including healthcare providers, in both acute and primary care settings, as well as the community stakeholders, as they play significant roles in monitoring and improving the vaccination rate and reducing the spread of contagious diseases such as the COVID-19 viral disease in the communities [1].

The inclusion of both healthcare professionals and stakeholders may be considered a strength of this community research study as it is important to examine and understand their views towards vaccination programs during the COVID-19 pandemic. Healthcare and community stakeholders have brought value to both pragmatic research and health service delivery. Furthermore, the analysis of data included in this study has been analyzed using relevant statistical analysis methods in addressing the study aims. There are five limitations in this study. The first limitation of this study is that the sample population mainly consisted of nurses and a few allied health professionals, which might not generalize the whole healthcare professionals’ knowledge, attitudes and behaviors towards the COVID-19 vaccine acceptance. A second limitation was that the online survey was only available in Cantonese and distributed via web platforms, hence decreasing the generalizability of findings to other non-Chinese speaking healthcare professionals and community stakeholders. A third study limitation is observed in the circumstances surrounding the ever-changing and rapidly evolving COVID-19 virus itself in conjunction with differing vaccine mandates. The fourth limitation is the potential overrepresentation of female participants in the study sample; our study results may be biased towards the female perspective. The fifth limitation is a common drawback of online surveys, whereby it is difficult to strictly verify the eligibility of the participants. Similar to other published online surveys, the validity of the study findings relies on the integrity of the participants’ self-reported responses. Further to this, study data were retrieved over a short segment of time (during wave two and wave three in Hong Kong). Future studies should focus on the well-thought strategies and approaches pertaining to vaccination campaigns and programs that healthcare professionals plan to implement to overcome perceived barriers to COVID-19 vaccinations. For future research, investigations regarding community stakeholders should look at investigating initiatives via community and mass media efforts to manage adverse events and inaccurate information. Vaccines are the fundamental tools to reduce the morbidity and mortality rates during the COVID-19 pandemic.

## 5. Conclusions

Vaccination is an essential approach for tackling the COVID-19 pandemic by reaching herd immunity in the population nationally and globally. This study identified that the knowledge and attitudes of healthcare professionals and community stakeholders are associated with high COVID-19 vaccine intent. These associated factors should be addressed by multilevel strategies to implement public health protocols and policies to fight against the COVID-19 pandemic locally, nationally and globally. It is important to understand the knowledge, attitudes and acceptance intention for the uptake of COVID-19 vaccination among healthcare professionals and community stakeholders in order to develop and plan relevant health education and health promotion for increasing herd immunity within the communities. Training and education regarding up-to-date information for healthcare providers and community stakeholders focusing on the delivery of evidence-based data on the benefits of vaccination programs for most of the populations around the world to increase the vaccination rates is recommended. This investigation, therefore, sets a stage towards the age of national information for offering interesting factors to implement public health protocols and policies to fight against the COVID-19 pandemic locally, nationally and globally.

## Figures and Tables

**Table 1 ijerph-19-14499-t001:** Background characteristics of participants (N = 512).

	N	%
Gender		
Male	83	16.2
Female	429	83.8
Age		
Below 40	272	53.1
40–59	198	38.7
60 or over	42	8.2
Healthcare related occupation		
Yes	297	58.0
No	215	42.0
Healthcare related organization		
Yes	258	50.4
No	254	49.6
Have you ever contacted those with higher risk of getting infected in your work and daily life?		
Yes	267	52.1
No	245	47.9
Have you family and friends ever admitted to hospital due to COVID-19 infection?		
Yes	40	7.8
No	472	92.2
Do you have chronic illnesses?		
Yes	58	11.3
No	454	88.7
Over the past three years, have you received flu vaccination?		
Yes	243	47.5
No	269	52.5

**Table 2 ijerph-19-14499-t002:** Frequency distributions of variables related to receive COVID-19 vaccination perceptions (N = 512).

	N	%
**Knowledge of COVID-19**		
** *Knowledge about COVID-19 transmission* **		
COVID-19 is highly contagious		
Yes *	501	97.9
No	7	1.4
Don’t know	4	0.8
Healthcare professional, older people and people with chronic illnesses are easier to get infected with COVID-19		
Yes *	471	92.0
No	36	7.0
Don’t know	5	1.0
Youths have higher risk of getting COVID-19		
Yes	61	11.9
No *	371	72.5
Don’t know	80	15.6
Asymptomatic people can transmit the viruses to others		
Yes	492	96.1
No	15	2.9
Don’t know	5	1.0
If you suffer from COVID-19, you could transmit the viruses to your family and friends		
Yes	503	98.2
No	1	0.2
Don’t know	8	1.6
Closed area and social gathering is the major ways of SARS-CoV-2 transmission		
Yes	488	95.3
No	15	2.9
Don’t know	9	1.8
The most effective ways to mitigate COVID-19 are hand hygiene, facial mask wearing and social distancing		
Yes	507	99.0
No	3	0.6
Don’t know	2	0.4
** *Knowledge about COVID-19* **		
You will have fever and sick for days if you suffer from COVID-19		
Yes	428	83.6
No	60	11.7
Don’t know	24	4.7
Your daily life will be affected if you suffer from COVID-19		
Yes	471	92.0
No	23	4.5
Don’t know	18	3.5
COVID-19 can cause serious and life-threatening complications in people with chronic illnesses		
Yes	501	97.9
No	7	1.4
Don’t know	4	0.8
COVID-19 may cause serious and life-threatening complications in children		
Yes	439	85.7
No	30	5.9
Don’t know	43	8.4
Majority of the young and healthy people would not have complications even they suffer from COVID-19		
Yes	204	39.8
No *	236	46.1
Don’t know	72	14.1
Currently there is no way to effectively cure COVID-19		
Yes	407	79.5
No	60	11.7
Don’t know	45	8.8
Those recovered from COVID-19 may suffer from pulmonary fibrosis		
Yes	454	88.7
No	9	1.8
Don’t know	49	9.6
Those recovered from COVID-19 may get infected with it again		
Yes	484	94.5
No	7	1.4
Don’t know	21	4.1
** *Knowledge about the effect of COVID-19 vaccine* **		
The vaccine can strengthen my immunity against COVID-19, so to reduce the chance of being infected with it		
Yes	397	77.5
No	31	6.1
Don’t know	84	16.4
The vaccine can lower the risk of transmitting the viruses to my family and friends		
Yes	360	70.3
No	71	13.9
Don’t know	81	15.8
The benefits of COVID-19 vaccination outweigh its harm		
Yes	300	58.6
No	47	9.2
Don’t know	165	32.2
Vaccination is an effective way to prevent COVID-19		
Yes	341	66.6
No	47	9.2
Don’t know	124	24.2
Vaccination can have 100% protection against COVID-19		
Yes	39	7.6
No *	409	79.9
Don’t know	64	12.5
There is a lifelong protection against COVID-19 after completing two doses of vaccine		
Yes	41	8.0
No *	354	69.1
Don’t know	117	22.9
People with history of severe allergy cannot have the COVID-19 vaccination		
Yes	357	69.7
No	46	9.0
Don’t know	109	21.3
**Negative Factors against COVID-19 vaccination**		
The vaccine is still under development. There is insufficient scientific evidence to prove its effectiveness against COVID-19		
Completely disagreed/disagreed	137	26.8
Completely agreed/agreed	375	73.2
The vaccine has unknown side effects which make you worrying about the safety of the vaccination		
Completely disagreed/disagreed	56	10.9
Completely agreed/agreed	456	89.1
The vaccine contains SARS-CoV-2 viruses. You may get infected from the vaccination		
Completely disagreed/disagreed	412	80.5
Completely agreed/agreed	100	19.5
You are not high-risk group. You can fully recover from the illness with no complications		
Completely disagreed/disagreed	385	75.2
Completely agreed/agreed	127	24.8
You have minimal chance of getting COVID-19 because you have done all the preventive measures		
Completely disagreed/disagreed	254	49.6
Completely agreed/agreed	258	50.4
The information regarding the vaccine is confusing, I don’t know which vaccine is suitable for me		
Completely disagreed/disagreed	157	30.7
Completely agreed/agreed	355	69.3
**Positive factors for COVID-19 vaccination**		
If more family members or friends get the vaccination without undesirable effects, I will consider the vaccination		
Completely disagreed/disagreed	154	30.1
Completely agreed/agreed	358	69.9
If the vaccination venue and time fits me better, it will increase my intention of vaccination		
Completely disagreed/disagreed	216	42.2
Completely agreed/agreed	296	57.8
It will encourage me to get the vaccination if the vaccination is recommended by prestigious healthcare professionals		
Completely disagreed/disagreed	213	41.6
Completely agreed/agreed	299	58.4
It will encourage me to get the vaccination if there is sufficient scientific evidence to support the effectiveness of the vaccines and show no severe side effects		
Completely disagreed/disagreed	51	10.0
Completely agreed/agreed	461	90.0
The general public has the responsibility to get the vaccination to reduce COVID-19 transmission		
Completely disagreed/disagreed	162	31.6
Completely agreed/agreed	350	68.4
The government should mandate all high-risk groups to get the vaccination		
Completely disagreed/disagreed	320	62.5
Completely agreed/agreed	192	37.5
The government and social media should provide accurate and clear information and instruction for the vaccine to the general public		
Completely disagreed/disagreed	51	10.0
Completely agreed/agreed	461	90.0
The government should set up a trust fund to provide assistance to those experiencing severe side effects after vaccination		
Completely disagreed/disagreed	51	10.0
Completely agreed/agreed	461	90.0
**Intention to receive COVID-19 vaccination**		
Will you receive COVID-19 vaccination?		
Must/high probability	333	65.0
Never/must not/low probability	179	35.0

* Appropriate response.

**Table 3 ijerph-19-14499-t003:** Associations between background variables ^#^ and the intention to get vaccinated (N = 512).

	Row %	OR_U_ (95%CI)	ORm (95%CI)
Age			
Below 40	52.6	1	1
40–59	77.3	3.067 (2.038, 4.616) ***	3.157 (2.090, 4.767) ***
60 or over	88.1	6.676 (2.547, 17.499) ***	6.606 (2.513, 17.360) ***
Over the past three years, have you received flu vaccination?			
Yes	69.5	1.462 (1.013, 2.110) *	1.537 (1.047, 2.258) *
No	61.0	1	1

* *p* < 0.05; *** *p* < 0.001. ^#^ Univariately non-significant variables, not considered in the model. OR_U_: univariate odds ratios. ORm: multivariate OR, odds ratios obtained from multivariate logistic analysis using background variables: age, gender, years of working experiences, received flu vaccines previously. 95% CI: 95% confidence interval.

**Table 4 ijerph-19-14499-t004:** Associations between factors related to COVID-19 vaccine and the intention to get vaccinated (N = 512).

	Row %	OR_U_ (95% CI)	AOR (95% CI)
** * Knowledge about COVID-19 transmissions * **			
**COVID-19 is highly contagious**			
Yes	65.3	1.566 (0.471, 5.205)	1.934 (0.534, 7.011)
No/Don’t know	54.5	1	1
**Healthcare professional, older people and people with chronic illnesses are easier to get infected with COVID-19**			
Yes	65.4	1.209 (0.628, 2.329)	1.055 (0.531, 2.096)
No/Don’t know	61.0	1	1
**Youths have higher risk of getting COVID-19**			
Yes	65.2	1.013 (0.674, 1.521)	1.081 (0.705, 1.657)
No/Don’t know	65.0	1	1
**Asymptomatic people can transmit the viruses to others**			
Yes	65.0	1.002 (0.392, 2.558)	1.060 (0.384, 2.930)
No/Don’t know	65.0	1	1
**If you suffer from COVID-19, you could transmit the viruses to your family and friends.**			
Yes	65.2	1.499 (0.398, 5.655)	1.594 (0.391, 6.503)
No/Don’t know	55.6	1	1
**Closed area and social gathering is the major ways of SARS-CoV-2 transmission**			
Yes	66.8	4.887 (1.987, 12.022) ***	4.688 (1.802, 12.199) ***
No/Don’t know	29.2	1	1
**The most effective ways to mitigate COVID-19 are hand hygiene, facial mask wearing and social distancing.**			
Yes	65.5	7.589 (0.842, 68.415)	8.886 (0.855, 92.359)
No/Don’t know	20.0	1	1
** * Knowledge about COVID-19 * **			
**You will have fever and sick for days if you suffer from COVID-19**			
Yes	64.3	0.806 (0.487, 1.332)	0.896 (0.527, 1.525)
No/Don’t know	69.0	1	1
**Your daily life will be affected if you suffer from COVID-19**			
Yes	64.5	0.753 (0.375, 1.515)	0.739 (0.356, 1.533)
No/Don’t know	70.7	1	1
**COVID-19 can cause serious and life-threatening complications in people with chronic illnesses.**			
Yes	65.3	1.566 (0.471, 5.205)	1.506 (0.430, 5.268)
No/Don’t know	54.5	1	1
**COVID-19 may cause serious and life-threatening complications in children**			
Yes	64.2	0.775 (0.453, 1.325)	1.100 (0.619, 1.954)
No/Don’t know	69.9	1	1
**Majority of the young and healthy people would not have complications even they suffer from COVID-19.**			
Yes	63.8	0.886 (0.615, 1.276)	0.837 (0.570, 1.228)
No/Don’t know	66.5	1	1
**Currently there is no way to effectively cure COVID-19.**			
Yes	65.1	1.015 (0.648, 1.591)	0.909 (0.566, 1.461)
No/Don’t know	64.8	1	1
**Those recovered from COVID-19 may suffer from pulmonary fibrosis**			
Yes	65.0	0.976 (0.550, 1.735)	1.241 (0.672, 2.291)
No/Don’t know	65.5	1	1
**Those recovered from COVID-19 may get infected with it again.**			
Yes	64.7	0.732 (0.316, 1.697)	0.753 (0.309, 1.830)
No/Don’t know	71.4	1	1
** * Knowledge about the effect of COVID-19 vaccine * **			
**The vaccine can strengthen my immunity against COVID-19, so to reduce the chance of being infected with it.**			
Yes	71.3	3.227 (2.103, 4.951) ***	2.983 (1.904, 4.674) ***
No/Don’t know	43.5	1	1
**The vaccine can lower the risk of transmitting the viruses to my family and friends.**			
Yes	71.4	2.495 (1.687, 3.691) ***	2.276 (1.508, 3.436) ***
No/Don’t know	50.0	1	1
**The benefits of COVID-19 vaccination outweigh its harm.**			
Yes	79.0	4.546 (3.085, 6.698) ***	3.913 (2.618, 5.847) ***
No/Don’t know	45.3	1	1
**Vaccination is an effective way to prevent COVID-19**			
Yes	76.2	4.309 (2.910, 6.380) ***	3.810 (2.535, 5.728) ***
No/Don’t know	42.7	1	1
**Vaccination can have 100% protection against COVID-19**			
Yes	63.1	0.900 (0.574, 1.410)	1.011 (0.632, 1.618)
No/Don’t know	65.5	1	1
**There is a lifelong protection against COVID-19 after completing two doses of vaccine**.			
Yes	65.2	1.010 (0.681, 1.496)	1.054 (0.698, 1.591)
No/Don’t know	65.0	1	1
**People with history of severe allergy cannot have the COVID-19 vaccination.**			
Yes	64.7	0.953 (0.641, 1.416)	0.953 (0.628, 1.446)
No/Don’t know	65.8	1	1
** * Factors against COVID-19 vaccination * **			
**The vaccine is still under development. There is insufficient scientific evidence to prove its effectiveness against COVID-19**			
Completely agreed/agreed	60.0	0.403 (0.255, 0.637) ***	0.481 (0.298, 0.775) ***
Completely disagreed/disagreed	78.8	1	1
**The vaccine has unknown side effects which make you worrying about the safety of the vaccination**			
Completely agreed/agreed	64.0	0.651 (0.350, 1.213)	0.681 (0.355, 1.307)
Completely disagreed/disagreed	73.2	1	1
**The vaccine contains SARS-CoV-2 viruses. You may get infected from the vaccination**			
Completely agreed/agreed	55.0	0.589 (0.378, 0.919) *	0.662 (0.414, 1.058)
Completely disagreed/disagreed	67.5	1	1
**You are not high-risk group. You can fully recover from the illness with no complications**			
Completely agreed/agreed	55.9	0.595 (0.395, 0.897) *	0.594 (0.385, 0.917) *
Completely disagreed/disagreed	68.1	1	1
**You have minimal chance of getting COVID-19 because you have done all the preventive measures**			
Completely agreed/agreed	58.5	0.558 (0.386, 0.807) ***	0.567 (0.386, 0.834) ***
Completely disagreed/disagreed	71.7	1	1
**The information regarding the vaccine is confusing, I don’t know which vaccine is suitable for me**			
Completely agreed/agreed	65.6	1.089 (0.735, 1.611)	1.306 (0.860, 1.983)
Completely disagreed/disagreed	63.7	1	1
** * Positive factors for COVID-19 vaccination * **			
**If more family members or friends get the vaccination without undesirable effects, I will consider the vaccination**			
Completely agreed/agreed	75.4	4.432 (2.967, 6.620) ***	3.714 (2.447, 5.636) ***
Completely disagreed/disagreed	40.9	1	1
**If the vaccination venue and time fits me better, it will increase my intention of vaccination**			
Completely agreed/agreed	80.4	5.226 (3.528, 7.744) ***	4.348 (2.895, 6.529) ***
Completely disagreed/disagreed	44.0	1	1
**It will encourage me to get the vaccination if the vaccination is recommended by prestigious healthcare professionals**			
Completely agreed/agreed	74.9	2.850 (1.959, 4.146) ***	2.377 (1.602, 3.528) ***
Completely disagreed/disagreed	51.2	1	1
**It will encourage me to get the vaccination if there is sufficient scientific evidence to support the effectiveness of the vaccines and show no severe side effects**			
Completely agreed/agreed	69.0	5.337 (2.832, 10.059) ***	4.653 (2.389, 9.063) ***
Completely disagreed/disagreed	29.4	1	1
**The general public has the responsibility to get the vaccination to reduce COVID-19 transmission**			
Completely agreed/agreed	78.0	6.027 (4.012, 9.055) ***	5.059 (3.309, 7.735) ***
Completely disagreed/disagreed	37.0	1	1
**The government should mandate all high-risk groups to get the vaccination**			
Completely agreed/agreed	76.6	2.353 (1.576, 3.514) ***	2.235 (1.467, 3.404) ***
Completely disagreed/disagreed	58.1	1	1
**The government and social media should provide accurate and clear information and instruction for the vaccine to the general public**			
Completely agreed/agreed	66.2	1.606 (0.895, 2.881)	1.531 (0.831, 2.823)
Completely disagreed/disagreed	54.9	1	1
**The government should set up a trust fund to provide assistance to those experiencing severe side effects after vaccination**			
Completely agreed/agreed	66.4	1.755 (0.980, 3.143)	1.858 (1.003, 3.440) *
Completely disagreed/disagreed	52.9	1	1

* *p* < 0.05; *** *p* < 0.001. AOR: adjusted OR, odds ratios after adjusting simultaneously for the variable and the significant background variables listed in Table 3. 95% CI: 95% confidence interval.

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
