# Peer review of "Factors Associated with COVID-19 Vaccine Acceptance among Healthcare Professionals and Community Stakeholders in Hong Kong: A Cross-Sectional Study"

_ijerph, 2022, doi:10.3390/ijerph192114499_

Round 1
Reviewer 1 Report
This is a very interesting paper exploring vaccine acceptance in healthcare and community stakeholders.
Minor suggestions:
- it should be made clearer that this research was conducted in Hong Kong, perhaps in the title and abstract, as this doesn't become apparent until the methods section.
- page 1-2 lines 44-45: it's not clear what are the barriers, do they refer to the factors at the beginning of the sentence? (The sentence just needs to be further clarified)
- Introduction section: there is now a significant body of research on acceptance of the vaccine, it would be beneficial to include more on how this specifically applies to the population of interest to this paper.
- Study design and participants: more clarity is needed particularly around community stakeholders, what is meant by this term, and which researcher made the decision about who to invite? Similar point in the results questions, what roles did they have for example?
- Page 3, line 103: is the footnote meant to be a reference?
- line 281, line 295, line 323: review the sentence to clarify.
- discussion: there is a significant amount of new research introduced in the discussion section which might be useful to include in the introduction section.
Author Response
Dear Reviewer No. 1,
Thank you for the reviewer's valuable comments. Please find the authors’ responses in the attached table.
The authors would like to express their gratitude for the reviewer’s time to provide your valuable comments.

Reviewer 2 Report
REVIEWER’S COMMENT
The paper entitled “Factors associated with COVID-19 Vaccine acceptance among healthcare professionals and community stakeholders: A cross- 3 sectional study” is an interesting and innovative study which has knowledge and attitudes of healthcare professionals and community stakeholders are associated with high COVID-19 vaccine intent.
It can offer interesting factors to implement public health protocols and policies to fight against the COVID-19 pandemic locally, nationally and globally. However, some critical observations may be outlined to be addressed and improved before final acceptance and publication by the journal:
1. The “Abstract” section: Here the authors enumerated the findings immediately after the very first sentence. Two or three lines could have been added on the nature of problem and factors associated.
2. The “Introduction” section: This section is well written and readers can be nicely introduced with the understanding of covid 19 cases worldwide and Acceptance of vaccination in both healthcare professionals and general public in the community.
3. The “Methods” section: the study was well designed with 520 healthcare professionals and stakeholders. But in results sections its written 512 were recruited this is confusing to me. Please explain.
4. The results can be explained with one or two graphs if Possible.
5. The “Discussion part” section: the study is looking interesting and have different results than previous study in Australia, Hong kong and Singapore which is very interesting but can the Author explain the possible reason for it.
6. The “Conclusion” section: The last line “This investigation is therefore a stage towards the age of national information” may be replaced with “This investigation, therefore, sets a stage towards the age of national information”. Also, one or two lines may be added here with a futuristic vision and the relevance of the present study towards that direction.
I strongly endorse the article for acceptance and publication in the journal, provided all the suggestions are taken care of and addressed by the authors meticulously, thereby improving the quality of the MS to provide a clear positive message to the intended readers globally.
Author Response
Dear Reviewer No. 2,
Thank you for the reviewer's valuable comments. Please find the authors’ responses in the attached table.
The authors would like to express their gratitude for the reviewer’s time to provide your valuable comments.

Reviewer 3 Report
Dear Authors, Attached is my report. Best regards,
Author Response
Dear Reviewer No. 3,
Thank you for the reviewer's valuable comments. Please find the authors’ responses in the attached table.
The authors would like to express their gratitude for the reviewer’s time to provide your valuable comments.
